

# Safety evaluation of four faba bean extracts used as dietary supplements in grass carp culture based on hematological indices, hepatopancreatic function and nutritional condition

Lingling Ma[1,2,*], Gen Kaneko[3,*], Jun Xie[1], Guangjun Wang[1], Zhifei Li[1], Jingjing Tian[1], Kai Zhang[1], Yun Xia[1], Wangbao Gong[1], Haihang Li[4] and Ermeng Yu[1]

[1] Key Laboratory of Tropical & Subtropical Fishery Resource Application & Cultivation, Pearl River Fisheries Research Institute of CAFS, Guangzhou, China
[2] National Demonstration Center for Experimental Fisheries Science Education, Shanghai Ocean University, Shanghai, China
[3] School of Arts & Sciences, University of Houston-Victoria, Victoria, TX, USA
[4] Guangdong Provincial Key Lab for Plant Development, School of Life Sciences, South China Normal University, Guangzhou, China
* These authors contributed equally to this work.

Corresponding author
Ermeng Yu, boyem34@hotmail.com

## ABSTRACT

Faba bean (*Vicia faba*, FB) is known to improve the texture of fish meat but retards growth possibly by inducing hemolysis, hepatopancreas damage, and metabolic disorder. In this study, we used ultrasonic processing to isolate four FB extracts (water extract, alcohol extract, proteins and residues) and examined their beneficial and detrimental effects. These extracts were separately mixed with commercial feed and fed to grass carp (*Ctenopharyngodon idellus*) using whole FB and commercial feed as controls. After fish were fed one of the six experimental diets for 50 d and 100 d, we evaluated the growth and hematological parameters, activities of metabolic enzymes, hepatopancreatic histology and oxidative response, and lipid metabolism. Results showed that both whole FB and FB residues caused growth retardation and hepatopancreas damage ($P < 0.05$), whereas growth performance was improved in the FB water and alcohol extract groups compared to the whole FB group. Although the FB water extract negatively affected the number and morphological parameters of red blood cells ($P < 0.05$), the hematological damage was less pronounced than that of the whole FB group. Excessive hepatopancreatic fat accumulation was found in the whole FB, FB alcohol extract and FB residues groups. Moreover, serious hepatopancreas damages were observed in the FB residues group. These results suggest that the beneficial and detrimental components of FB were successfully separated in the four extracts, and the FB water extract would be the best choice for grass carp culture in terms of growth performance and health. The safety evaluation of the four FB extracts would facilitate further application of FB in aquatic feed.

# INTRODUCTION

Crisp grass carp (*Ctenopharyngodon idellus* C.et V), a high-quality grass carp fed solely on faba bean (*Vicia faba*) for 90–120 days, has been protected as the "China Geographical Indication Product" and its fillet products are exported to North America (*Ma et al., 2020*). Compared with ordinary grass carp, crisp grass carp has unique muscle qualities, such as increased hardness and springiness (*Lin, Zeng & Zhu, 2009*; *Xu et al., 2020*; *Yu et al., 2020*). As a potential protein source, faba bean has also been used to partially replace animal protein sources in human diets (*Multari, Stewart & Russell, 2015*). However, faba bean is known to cause growth retardation and organ damages in fish (*Li et al., 2018*; *Macarulla et al., 2001*; *Tian et al., 2019*; *Zhang et al., 2015*), thus requiring further research on its beneficial and detrimental effects.

Many previous studies have shown that long-term or excessive intake of faba bean has various detrimental effects on fish such as hemolysis, hepatopancreas damage, and metabolic disorders. The blockage of the blood circulation system is one of the main negative effects of faba bean feeding in grass carp culture (*Tan & Li, 2005*), and severe damages to the intestinal tract, head kidney, hepatopancreas, and serum antioxidant enzymes have also been reported in this species (*Gan et al., 2017*; *Chen et al., 2019*). The hepatopancreatic steatosis caused by long-term ingestion of faba bean may be related to the increased de novo synthesis of fatty acids, enhanced oxidative stress, and mitochondrial damage (*Fu et al., 2020*). Although a certain level of protein concentrate from faba bean can be efficiently utilized as a dietary protein source for Atlantic salmon (*Salmo salar*) (*De Santis et al., 2016*), diets containing more than 10% of faba bean have adverse effects on growth performance, hematological parameters, and serum biochemical parameters in the beluga (*Huso huso*) juveniles (*Soltanzadeh et al., 2016*). However, it is unclear which components of faba bean are responsible for these organ damages.

Our previous study isolated four types of faba bean extracts—FB water extract, FB alcohol extract, FB proteins, and FB residues—and examined their effects on the muscle quality of grass carp including increased hardness, collagen contents, and fiber density (*Ma et al., 2020*). The aim of the present study is to test the effects of these four FB extracts on the growth, hematology, hepatopancreatic function, and lipid metabolism of grass carp, in a 100-d feeding trial using six diets including commercial feed, whole faba bean (FB), and the four FB extract feeds (FB water extract feed, FB alcohol extract feed, FB proteins feed, and FB residues feed).

# MATERIALS AND METHODS

## Experimental feeds

Four FB extracts (FB water extract, FB alcohol extract, FB proteins, and FB residues) were obtained through methods in our previous study (*Ma et al., 2020*). In short, shelled FB was

**Table 1 Nutritive components of four faba bean extracts and six diets.**

| | Crude protein (g/100 g) | Crude fat (g/100 g) | Moisture (g/100 g) | Crude ash (g/100 g) |
|---|---|---|---|---|
| FBA | 36.8 | 10.5 | 10.4 | <0.1 |
| FBW | 37.4 | <0.5 | 10.2 | 3.9 |
| FBP | 58.6 | 2.8 | 10.1 | 3.9 |
| FBR | 6.6 | 0.65 | 12.2 | 7.7 |
| Control diet | 28.8 | 5.5 | 10.9 | 8.1 |
| FB diet | 28 | 1.4 | 14.4 | 4.1 |
| FBA diet | 36.8 | 10.5 | 10.4 | 8.05 |
| FBW diet | 29.85 | 4.87 | 10.81 | 7.75 |
| FBP diet | 37.71 | 4.69 | 10.66 | 6.84 |
| FBR diet | 17.72 | 3.08 | 11.55 | 7.9 |

Note:
Control, commercial feed; FB, whole faba bean; FBA, FB alcohol extract; FBW, FB water extract; FBP, FB proteins; FBR, FB residues.

ground into 60-um powder, mixed with distilled water, and adjusted pH to 9.0. After ultrasonic processing and followed by centrifugation, the FB water extract was obtained by concentrating the water supernatant. The precipitates were dried, and the FB proteins were thus obtained. After the residues from the first centrifugation were resuspended in ethanol, the FB alcohol extract was obtained by concentrating the ethanol extract, with the remaining precipitate representing the FB residues. To maintain the activities of heat-sensitive components, moderate temperature (<55 °C) was used throughout the extracting and formulating processes. In addition to the four experimental feeds (each weighing 16 kg) including FB water extract feed, FB alcohol extract feed, FB proteins feed, and FB residues feed, we used whole FB and commercial feed as controls (Ma et al., 2020). The nutritional components of these extracts and controls are shown in Table 1.

## Fish culture

Grass carp with a body weight of 750 ± 36 g were randomly allocated into six groups with two tanks per group and a density of eight individuals per tank. The fish were fed one of the six diets (four FB extracts feeds, commercial feed, or whole FB) at 9:00 am and 4:00 pm each day for 100 days, and the feed amount for each day was 1–2% of fish weight. The water temperature was kept at 25–30 °C, pH was 6.5–7.5, and dissolved oxygen (DO) was above 5.0 mg/L.

## Sampling procedures

On 50 d and 100 d, three grass carp were sampled from each group. The fish were anesthetized with pH-buffered tricaine methanesulfonate (250 mg/L), and 8 mL of blood per fish was collected from the vertebral blood vessel. Three mL of blood was placed in an anticoagulant vessel containing EDTAK2 and used for hematologic analyses with an automatic blood cell analyzer (BC-5000; Shenzhen Mindray Bio-medical Electronics Co. Ltd., Shenzhen, China). After standing for 4 h, serum was separated from the remaining

5 mL of blood by centrifugation at 3,500 r/min for 10 min and immediately stored at −80 °C for further biochemical analyses. Next, the body weight, body length, and the weight of the viscera, hepatopancreas, and mesenteric fat were measured to determine growth performance. The hepatopancreas (2 mm$^3$) was fixed in Bouin's reagent for 18 h and used for hematoxylin and eosin (H&E) staining. A small piece of the hepatopancreas was also sampled for enzymatic measurements.

The experimental protocols used in this study were approved by the Animal Ethics Committee of the Guangdong Provincial Zoological Society, China, under permit number GSZ-AW012.

## Blood biochemical analyses

The serum lipid metabolism was tested by measuring the total cholesterol (TC), triglyceride (TG), high density lipoprotein cholesterol (HDL-C), and low density lipoprotein cholesterol (LDL-C) levels using an automatic biochemical analyzer (7020; Hitachi high-tech, Tokyo, Japan). The activities of the serum enzymes, including glutamic oxalacetic transaminase (GOT), glutamic-pyruvic transaminase (GPT), alkaline phosphatase (AKP) and peroxidase (POD), were also measured by using GOT, GPT, AKP and POD reagent kits (Jiancheng, Nanjing, China), respectively. Absorbance was detected according to our previous methods (Ma et al., 2020).

## Hepatopancreatic enzyme activity measure

The hepatopancreatic lipid metabolism was assessed by measuring TC and TG levels and the activities of GPT, GOT, AKP and POD. Antioxidant parameters including superoxide dismutase (SOD), total glutathione (T-GSH), oxidative glutathione (GSSH), hydrogen peroxide ($H_2O_2$), nicotinamide adenine dinucleotide (NADPH), and POD were also measured by our previous methods (Ma et al., 2020). The hepatopancreas was homogenized in nine volumes of saline per gram using the Tissuelyser-24 automatic sample rapid grinding machine (Jingxing Instruments, Shanghai, China), centrifuged at 3,500 r/min for 10 min at 4 °C, and the resulting supernatant was used for the analysis. To measure T-GSH and GSSH, four volumes of saline were used to prepare the hepatopancreatic homogenate.

## Hepatopancreatic histology

Hematoxylin and eosin (H&E) staining of hepatopancreas samples were performed as described previously (Ma et al., 2020). The stained slides of hepatopancreas samples from different groups were observed for histological analyses.

## Data analyses

The data were analyzed using the SPSS 23 software (SPSS Inc., Chicago, IL, USA). The results are shown as "Mean ± SE." All the assessed variables were subjected to the analysis of variance (ANOVA) followed by Duncan's test to determine the existence of significant differences between groups ($P < 0.05$).

Table 2 Effect of faba bean and four extracts on growth parameters of grass carp on 50 d and 100 d.

| Days | Groups | Weight gain rate (%) | Condition factor (%) | Visceral somatic index (%) | Hepatopancreas somatic index (%) | Abdominal fat index (%) |
|---|---|---|---|---|---|---|
| 50 d | Control | 27.42 ± 1.37[bc] | 20.72 ± 0.67[ab] | 11.13 ± 0.67[ab] | 1.97 ± 0.08[a] | 1.68 ± 0.38[ab] |
| | FB | 14.24 ± 0.48[d] | 21.26 ± 2.04[ab] | 11.05 ± 1.09[ab] | 1.48 ± 0.10[a] | 2.01 ± 0.39[a] |
| | FBA | 18.7 ± 0.60[cd] | 20.90 ± 1.38[ab] | 9.37 ± 0.51[b] | 1.96 ± 0.23[a] | 1.14 ± 0.22[ab] |
| | FBW | 43.34 ± 3.55[a] | 24.34 ± 1.59[a] | 14.86 ± 3.02[a] | 1.93 ± 0.17[a] | 1.26 ± 0.22[ab] |
| | FBP | 33.11 ± 2.96[b] | 22.73 ± 0.70[ab] | 9.09 ± 0.77[b] | 1.83 ± 0.28[a] | 1.55 ± 0.31[ab] |
| | FBR | 2.96 ± 0.41[e] | 18.38 ± 1.78[b] | 4.62 ± 0.61[c] | 0.64 ± 0.14[b] | 0.84 ± 0.08[b] |
| 100 d | Control | 34.48 ± 2.18[b] | 35.58 ± 4.16[a] | 12.93 ± 0.28[a] | 3.23 ± 0.25[a] | 2.62 ± 0.05[a] |
| | FB | 8.19 ± 0.39[d] | 27.16 ± 0.59[b] | 13.07 ± 1.91[a] | 1.83 ± 0.18[c] | 3.66 ± 0.63[a] |
| | FBA | 47.33 ± 3.29[a] | 40.80 ± 3.08[a] | 13.92 ± 0.32[a] | 3.08 ± 0.19[ab] | 3.24 ± 0.68[a] |
| | FBW | 40.12 ± 1.72[ab] | 39.27 ± 2.27[a] | 12.53 ± 1.36[a] | 3.14 ± 0.36[ab] | 3.17 ± 0.65[a] |
| | FBP | 38.27 ± 0.09[b] | 35.01 ± 1.03[a] | 11.90 ± 0.53[a] | 2.31 ± 0.34[bc] | 3.87 ± 0.53[a] |
| | FBR | 10.91 ± 1.05[d] | 27.46 ± 0.68[b] | 10.50 ± 0.86[a] | 2.53 ± 0.13[abc] | 2.37 ± 0.30[a] |

Note:
Control, commercial feed; FB, whole faba bean; FBA, FB alcohol extract; FBW, FB water extract; FBP, FB proteins; FBR, FB residues. Values of the same column with different letters (a–e) are significantly different ($P < 0.05$).

# RESULTS

## Growth parameters

The growth parameters in different groups are shown in Table 2. On 50 d, the weight gain rates (WGR) of the FB water extract group was higher than that of the control group ($P < 0.05$), while the WGR, visceral somatic indexes (VSI), and hepatopancreas somatic index were the lowest in the FB residues group ($P < 0.05$). On 100 d, the WGR of the FB water extract group was higher compared with the control, but the WGR of the whole FB and FB residues groups were the lowest ($P < 0.05$). These two groups also displayed a lower condition factor. The VSI and abdominal fat index tended to fluctuate with some significant differences on 50 d, but there was no significant difference for the two parameters on 100 d ($P > 0.05$).

## Hematological analyses

To evaluate the morphology and function of blood cells, we chose several parameters, shown in Table 3, such as red blood cell count (RBC), hematocrit (HCT), hemoglobin (HGB), and white blood cell count (WBC) (see "Discussion" for rationale). On 50 d, compared with the control group, only the FB water extract group showed a slightly lower level in RBC. Other three FB extracts and whole FB groups did not show significant difference in RBC. The FB and FB water extract groups had lower levels of RBC, HCT and HGB compared with the FB alcohol extract group on 100 d ($P < 0.05$). There were no significant differences in the WBC among these groups ($P > 0.05$). In the FB residues group, the number (NEU#) and percentage (NEU%) of neutrophils were higher than those of the other groups on 100 d ($P < 0.05$). There were no significant differences in other hematological parameters among the six groups ($P > 0.05$).

**Table 3 Effect of the faba bean and four faba bean extracts on hematological parameters of grass carp on 50 d and 100 d.**

| Groups | 50 d | | | | | | 100 d | | | | | |
|---|---|---|---|---|---|---|---|---|---|---|---|---|
| | Control | FB | FBA | FBW | FBP | FBR | Control | FB | FBA | FBW | FBP | FBR |
| RBC ($10^{12}$/L) | 2.64 ± 0.38[a] | 1.98 ± 0.45[bc] | 2.34 ± 0.16[ab] | 2.58 ± 0.25[a] | 2.54 ± 0.20[a] | 1.73 ± 0.14[c] | 2.37 ± 0.29[ab] | 2.15 ± 0.10[bc] | 2.47 ± 0.04[a] | 2.00 ± 0.07[c] | 2.42 ± 0.22[ab] | 2.21 ± 0.11[abc] |
| HCT (%) | 37.63 ± 6.54[ab] | 29.37 ± 1.02[bc] | 36.00 ± 4.27[abc] | 37.43 ± 3.35[ab] | 46.83 ± 13.06[a] | 24.70 ± 2.78[c] | 32.40 ± 4.09[ab] | 29.23 ± 1.43[b] | 43.23 ± 15.17[a] | 28.80 ± 1.45[b] | 31.30 ± 4.10[ab] | 31.63 ± 0.83[ab] |
| MCV (fL) | 142.13 ± 4.85[a] | 155.20 ± 44.38[a] | 155.37 ± 29.73[a] | 145.53 ± 6.16[a] | 182.97 ± 41.90[a] | 142.30 ± 4.59[a] | 136.90 ± 3.72[a] | 135.97 ± 7.21[a] | 175.47 ± 63.23[a] | 143.67 ± 2.56[a] | 129.40 ± 4.60[a] | 143.10 ± 3.92[a] |
| HGB (g/L) | 123.67 ± 19.63[a] | 92.33 ± 16.26[bc] | 107.33 ± 10.69[ab] | 118.67 ± 15.95[a] | 120.00 ± 11.14[a] | 77.00 ± 7.00[c] | 108.00 ± 14.53[ab] | 97.00 ± 5.57[b] | 116.33 ± 5.51[a] | 93.00 ± 4.58[b] | 100.00 ± 13.00[ab] | 104.00 ± 2.65[ab] |
| MCH (pg) | 46.73 ± 0.91[a] | 47.07 ± 3.59[a] | 45.80 ± 1.65[a] | 46.10 ± 2.46[a] | 47.17 ± 2.01[a] | 44.27 ± 0.70[a] | 45.60 ± 0.87[a] | 45.13 ± 0.49[a] | 47.20 ± 2.46[a] | 46.33 ± 1.07[a] | 41.30 ± 1.40[b] | 46.90 ± 1.11[a] |
| WBC ($10^9$/L) | 17.76 ± 7.17[a] | 8.82 ± 4.40[b] | 12.53 ± 2.45[ab] | 13.58 ± 2.67[ab] | 12.86 ± 0.53[ab] | 11.05 ± 3.25a[b] | 16.85 ± 2.90[a] | 11.34 ± 1.94[a] | 15.67 ± 5.06[a] | 17.75 ± 4.25[a] | 17.40±0.22[a] | 16.51 ± 2.35[a] |
| NEU# ($10^9$/L) | 5.94 ± 5.24[a] | 0.49 ± 0.75[b] | 0.22 ± 0.11[b] | 0.12 ± 0.09[b] | 0.15 ± 0.10[b] | 0.08 ± 0.07[b] | 0.90 ± 0.61[b] | 0.78 ± 0.64[b] | 1.32 ± 0.55[b] | 2.25 ± 2.10[b] | 0.26 ± 1.81[b] | 6.49 ± 2.17[a] |
| NEU% | 28.27 ± 22.33[a] | 5.30 ± 7.72[b] | 1.90 ± 1.15[b] | 0.87 ± 0.57[b] | 1.13 ± 0.75[b] | 0.63 ± 0.38[b] | 5.03 ± 2.83[b] | 7.67 ± 7.66[b] | 8.90 ± 4.62[b] | 12.17 ± 11.33[b] | 12.10 ± 1.50[b] | 38.57 ± 7.77[a] |
| LY ($10^9$/L) | 10.84 ± 1.25[a] | 8.02 ± 4.30[a] | 11.81 ± 3.02[a] | 13.07 ± 2.49[a] | 12.40 ± 0.73[a] | 10.25 ± 2.96[a] | 15.63 ± 2.05[a] | 9.97 ± 2.70[ab] | 13.77 ± 4.58[ab] | 14.88 ± 4.29[ab] | 16.34 ± 3.80[a] | 8.70 ± 0.83[b] |
| LY% | 67.23 ± 24.85[b] | 90.40 ± 6.74[a] | 93.57 ± 5.82[a] | 96.33 ± 2.02[a] | 96.40 ± 2.17[a] | 92.87 ± 1.66[a] | 93.23 ± 4.31[a] | 86.83 ± 10.34[a] | 87.63 ± 5.26[a] | 84.20 ± 13.70[a] | 93.90 ± 11.20[a] | 53.87 ± 12.86[b] |
| EOS# ($10^9$/L) | 0.98 ± 1.14[a] | 0.32 ± 0.08[a] | 0.50 ± 0.49[a] | 0.39 ± 0.34[a] | 0.31 ± 0.32[a] | 0.72 ± 0.26[a] | 0.33 ± 0.31[a] | 0.59 ± 0.32[a] | 0.58 ± 0.39[a] | 0.62 ± 0.50[a] | 0.80 ± 0.23[a] | 1.32 ± 1.06[a] |
| PLT ($10^9$/L) | 12.67 ± 5.03[bc] | 24.00 ± 11.00[ab] | 19.00 ± 6.00[abc] | 7.33 ± 5.57[c] | 13.00 ± 5.57[bc] | 29.00 ± 13.75[a] | 25.00 ± 12.17[ab] | 14.00 ± 4.36[b] | 27.33 ± 4.62[a] | 25.33 ± 4.04[ab] | 15.00 ± 1.00[b] | 18.67 ± 3.51[ab] |
| PDW (%) | 17.50 ± 1.27[ab] | 18.67 ± 0.35[ab] | 18.40 ± 1.01[ab] | 17.00 ± 1.04[b] | 18.27 ± 0.78[ab] | 19.00 ± 0.35[a] | 17.87 ± 0.59[ab] | 17.83 ± 0.55[b] | 18.57 ± 0.45[ab] | 18.17 ± 0.25[ab] | 18.00 ± 0.00[ab] | 18.63 ± 0.25[a] |
| MPV (fL) | 7.97 ± 1.00[ab] | 8.97 ± 0.45[ab] | 8.60 ± 0.78[ab] | 7.70 ± 0.98[b] | 7.90 ± 0.30[ab] | 9.47 ± 1.12[a] | 8.43 ± 0.99[a] | 8.23 ± 1.00[a] | 8.73 ± 0.58[a] | 8.63 ± 0.40[a] | 8.60 ± 0.10[a] | 8.73 ± 0.40[a] |
| PCT (%) | 0.01 ± 0.00[bc] | 0.02 ± 0.01[ab] | 0.02 ± 0.00[abc] | 0.01 ± 0.00[c] | 0.01 ± 0.00b[c] | 0.03 ± 0.01[a] | 0.02 ± 0.01[ab] | 0.01 ± 0.00[b] | 0.02 ± 0.01[a] | 0.02 ± 0.00[ab] | 0.01 ± 0.00[ab] | 0.02 ± 0.00[ab] |

**Notes:**

Control, commercial feed; FB, whole faba beans; FBA, FB alcohol extract; FBW, FB water extract; FBP, FB proteins; FBR, FB residues.
RBC ($10^{12}$/L), red blood cell count; HCT (%), hematocrit; MCV (fL), mean corpuscular volume; HGB (g/L), hemoglobin; MCH (pg), mean erythrocyte hemoglobin contents; WBC ($10^9$/L), white blood cell count; NEU#, number of neutrophils; NEU%, percentage of neutrophils; LY ($10^9$/L), lymphocyte count; EOS# ($10^9$/L), eosinophil number; PLT ($10^9$/L), Platelet count; PDW (%), platelet distribution width; MVP (fL), mean platelet volume; PCT (%), plateletecrit. Values of the same row with different letters (a–c) are significantly different ($P < 0.05$).

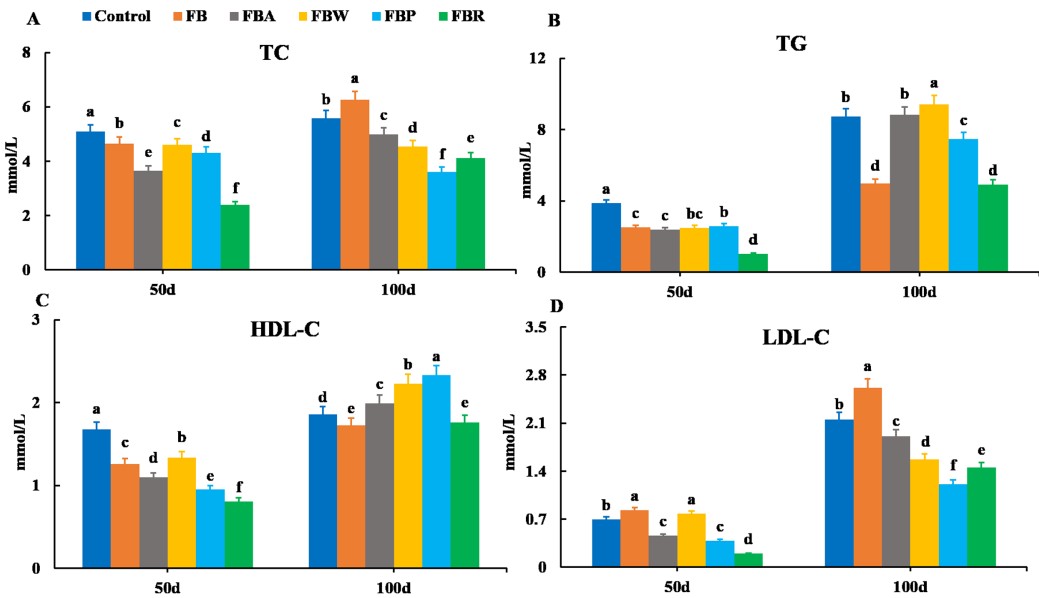

**Figure 1 Serum lipid metabolism parameters of grass carp on 50 d and 100 d.** Control, commercial feed; FB, whole faba bean; FBA, FB alcohol extract; FBW, FB water extract; FBP, FB proteins; FBR, FB residues. (A) Total cholesterol (TC); (B) triglycerides (TG); (C) high density lipoprotein cholesterol (HDL-C); (D) low density lipoprotein cholesterol (LDL-C). Different letters are significantly different ($P < 0.05$).

## Serum lipid and nutritional parameters

Serum lipid metabolism parameters are shown in Fig. 1. On 50 d, the TC, TG, and HDL-C contents were the highest in the control group and the lowest in the FB residues group ($P < 0.05$) (Figs. 1A–1C). In terms of LDL-C content, the FB and FB water extract groups showed the highest levels, while the FB residues group showed the lowest ($P < 0.05$) (Fig. 1D). On 100 d, both the TG and HDL-C contents were the lowest in the FB and FB residues groups ($P < 0.05$) (Figs. 1B and 1D). Both the TC and LDL-C contents of the whole FB group were higher than those of other groups on 100 d. The four FB extract groups had significantly lower levels of TC and LDL-C than the control group ($P < 0.05$) (Figs. 1A and 1C).

Changes in serum enzymes are shown in Fig. 2. There were no significant differences in the GOT and AKP activities among groups on 50 d ($P > 0.05$). However, on 100 d, the GOT activities of the FB and FB protein groups were lower than those of the control and whole FB groups, while the AKP activities of the FB alcohol extract, water extract and protein groups were higher than those of the control group ($P < 0.05$). On 50 d, the GPT activity was the highest in the FB protein group and then in the FB alcohol extract group ($P < 0.05$), while on 100 d it was the highest in the FB alcohol extract group ($P < 0.05$) (Fig. 2A). The POD activity of the FB proteins group was higher than those of the other groups on 50 d and 100 d, respectively ($P < 0.05$) (Figs. 2B–2D).

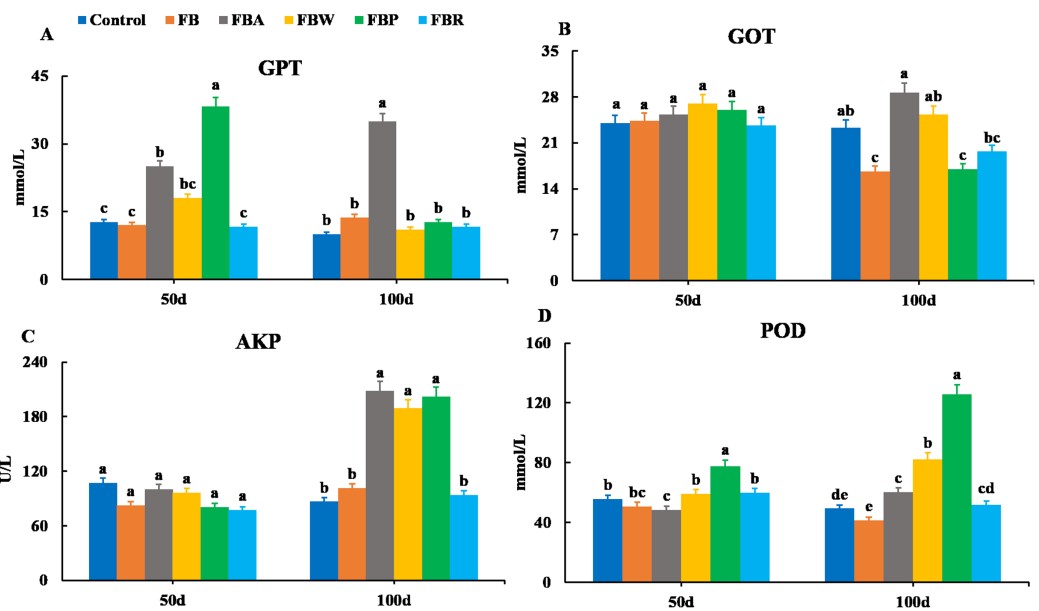

**Figure 2 Serum enzymatic parameters of grass carp on 50 d and 100 d.** Control, commercial feed; FB, whole faba bean; FBA, FB alcohol extract; FBW, FB water extract; FBP, FB proteins; FBR, FB residues. (A) Glutamic-pyruvic transaminase (GPT); (B), glutamic oxalacetic transaminase (GOT); (C) alkaline phosphatase (AKP); (D) peroxidase (POD). Different letters are significantly different ($P < 0.05$).

## Hepatopancreatic lipid and metabolic parameters

The TC and TG contents were also observed in the hepatopancreas along with the AKP, GOT, GPT and POD activities (Fig. 3). On 100 d, the TC contents of the FB water and FB alcohol groups were higher than those of the other groups (Fig. 3A), while the TG contents of the whole FB group were also higher than those of other groups ($P < 0.05$) (Fig. 3B). This phenomenon might be related to the normal developmental process where grass carp starts to accumulate visceral fat around 60 d (*Tian et al., 2019*). The AKP activity of whole FB group was the highest among the six groups both on 50 d and 100 d ($P < 0.05$) (Fig. 3C). The whole FB group showed the higher value of GPT activity (Fig. 3E), but GOT and POD activities of whole FB group were lower than those of the control on 100 d (Figs. 3D and 3F).

## Hepatopancreatic oxidative abilities

Changes in the oxidative parameters of SOD, NADPH, NADP$^+$, H$_2$O$_2$, T-GSH, and GSSH were measured in the hepatopancreas (Fig. 4). On 100 d, the SOD activity, NADPH content, and H$_2$O$_2$ content of the FB and four FB extract groups were lower than those of the control group ($P < 0.05$) except for the H$_2$O$_2$ content of the FB water extract group (Figs. 4A, 4B and 4D). The T-GSH and GSSH contents of the FB, FB alcohol extract and FB residues groups were higher than those of the control group on 100 d ($P < 0.05$) (Figs. 4E and 4F). The NADP$^+$ content of the FB, FB proteins and FB residues groups were higher than that of the control group on 100 d ($P < 0.05$) (Fig. 4C).

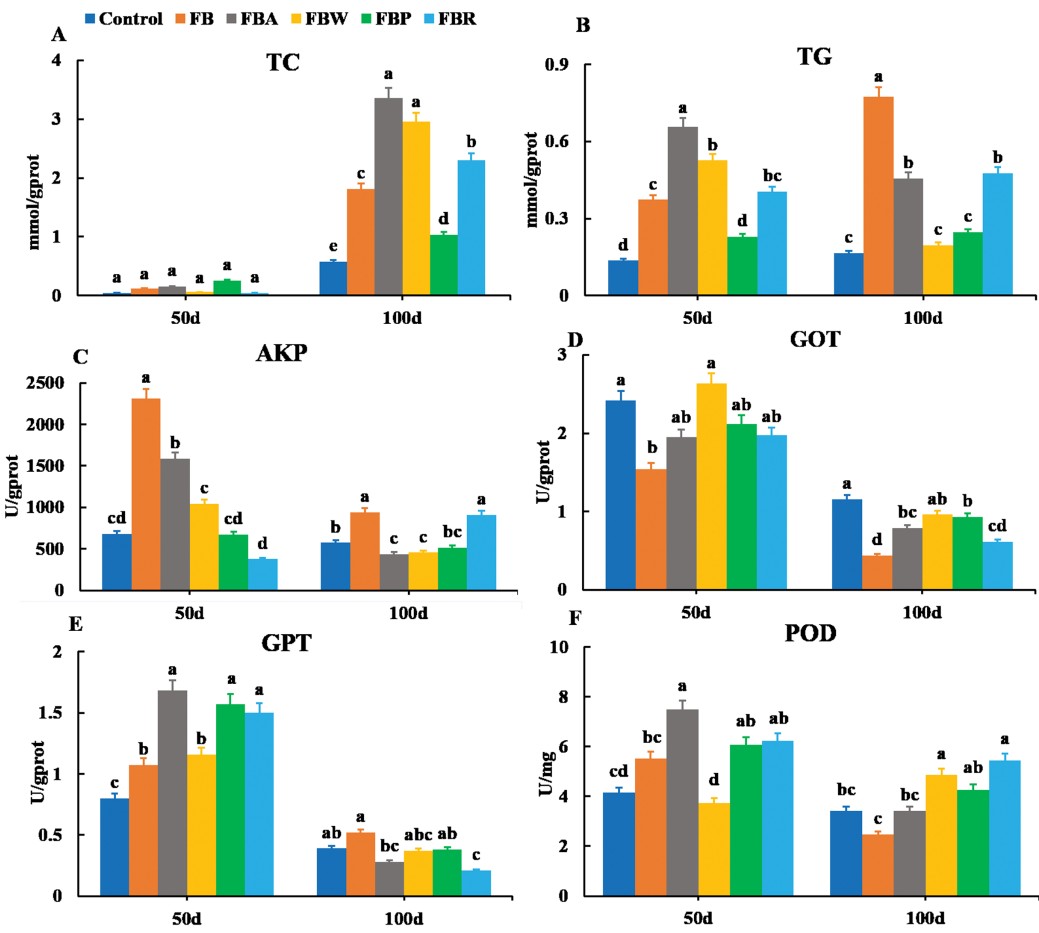

**Figure 3 Hepatopancreatic lipid metabolic parameters of grass carp on 50 d and 100 d.** Control, commercial feed; FB, whole faba bean; FBA, FB alcohol extract; FBW, FB water extract; FBP, FB proteins; FBR, FB residues. (A) Total cholesterol (TC); (B) triglycerides (TG); (C) alkaline phosphatase (AKP); (D) glutamic-pyruvic transaminase (GPT); (E) glutamic oxalacetic transaminase (GOT); (F) peroxidase (POD). Different letters are significantly different ($P < 0.05$).

## Histological structure of hepatopancreas

The hepatopancreas microstructures of the six groups were observed on 100 d (Fig. 5). Most lipid vacuoles (LV) were found in the FB and FB alcohol extract groups (Figs. 5B and 5C), followed by the FB water extract and FB proteins groups. Compared with the control group, the number of nuclei (N) were reduced in the FB water extract and FB proteins groups (Figs. 5A, 5D and 5E). In addition, minor cellular infiltration was found in the FB proteins and FB residues groups (Figs. 5E and 5F).

## DISCUSSION

Hematological analysis is an important tool in evaluating the physiological status of fish since hematological parameters are affected by a multitude of intrinsic and extrinsic factors, such as feed, health condition, oxygen, etc. (*Clauss, Dove & Arnold, 2008*; *Fazio, 2018*). It has been reported that ingestion of faba bean leads to hemolysis, which results in

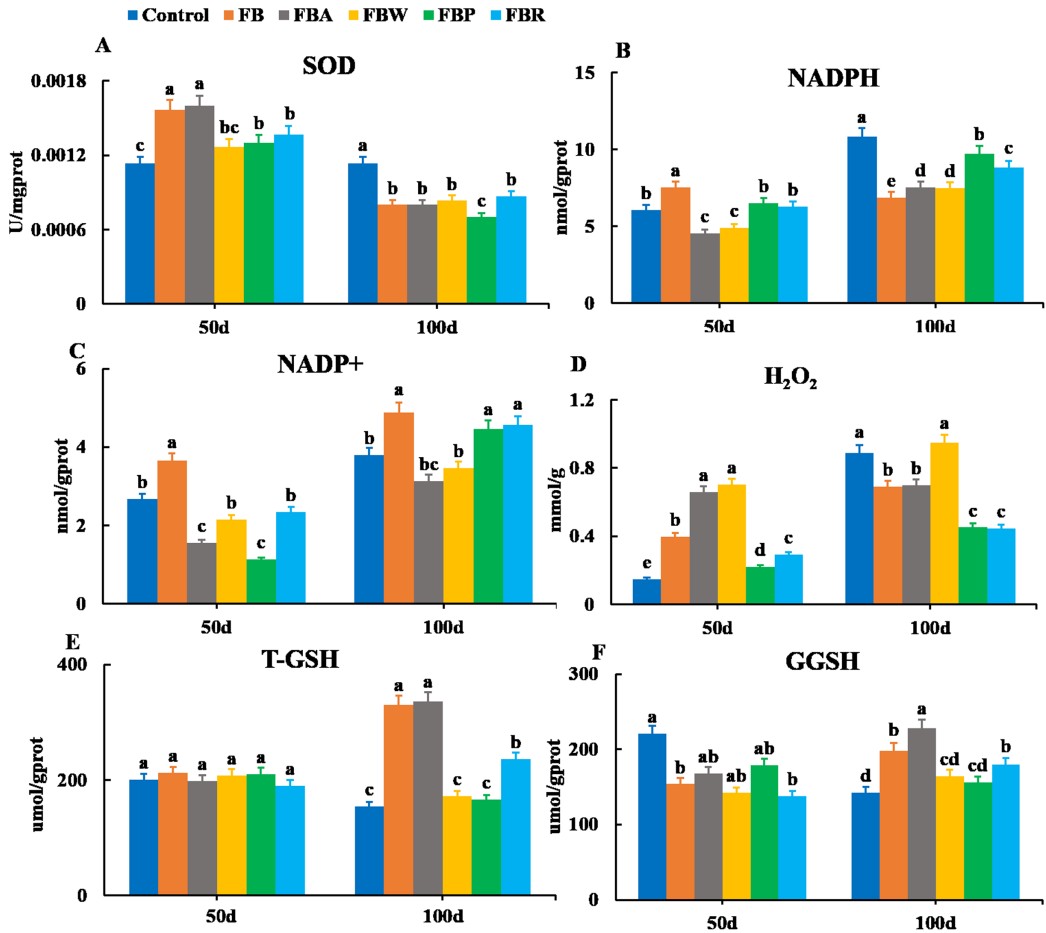

**Figure 4 Hepatopancreatic oxidative enzymes of grass carp on 50 d and 100 d.** Control, commercial feed; FB, whole faba bean; FBA, FB alcohol extract; FBW, FB water extract; FBP, FB proteins; FBR, FB residues. (A) Superoxide dismutase (SOD); (B) nicotinamide adenine dinucleotide (NADPH); (C) nicotinamide adenine dinucleotide phosphate (NADP$^+$); (D) photohydrogen peroxide (H$_2$O$_2$); (E) total glutathione (T-GSH); (F) oxidative glutathione (GGSH). Different letters are significantly different ($P < 0.05$).       

reduced red blood cell count (RBC) and decreased hemoglobin content, and thereby increases the oxygen requirement in grass carp (*Yu et al., 2017*). Our previous study has found that the FB and FB water extract groups have low RBC and glucose-6-phosphate dehydrogenase (G6PD) activity, which possibly causes hemolysis (*Ma et al., 2020*). In this study, we conducted a more detailed hematological analysis. The results showed evidence of oxidative damage to red blood cells, but the FB water extract group displayed improved count and morphology compared to the whole FB group on 50 d. These results suggest that the acute damage caused by FB could be alleviated in the FB water extract group compared to the whole FB group. White blood cell parameters, on the other hand, are generally associated with pathological conditions of fish. Because our experimental fish appeared healthy throughout the feeding trial, it is not surprising that we observed no clear tendencies in white blood cell parameters. Only the group fed with

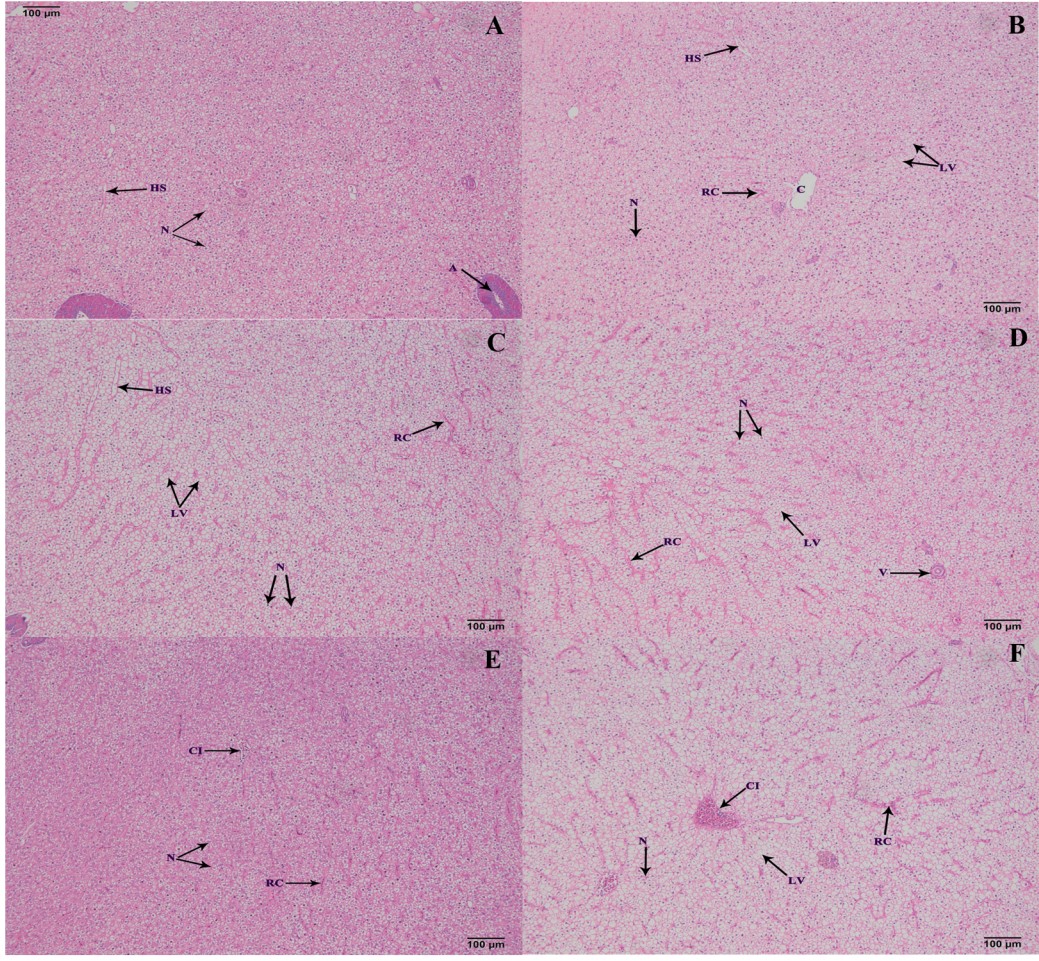

**Figure 5 Histological structure of hepatic tracts from grass carp on 100 d.** (A) Control, commercial feed; (B) FB, whole faba bean; (C) FBA, FB alcohol extract; (D) FBW, FB water extract; (E) FBP, FB proteins; (F) FBR, FB residues. H&E, bar = 100 µm. Lipid vacuole (LV), central vein (C), nucleus (N), hepatic sinusoid (HS), red cell (RC) and cellular infiltration (CI).

the FB residues feed, which appears to be the most detrimental, showed a marked increase in neutrophils.

Lipid metabolism is related to fish nutrition and health, and the dysfunction of lipid metabolism (dyslipidemia) often leads to fat accumulation, which mainly results in high total cholesterol (TC) and high triglycerides (TG) contents (*Howard, Ruotolo & Robbins, 2003*). Previous studies have shown that fat accumulation increases in grass carp that are fed faba bean (*Tian et al., 2019*; *Wang et al., 2015*). The results of the present study further suggest that the causative compound(s) of hepatopancreatic fat accumulation is/are alcohol-soluble for the following reasons. First, high TC and TG contents were observed in the hepatopancreas of both the FB and FB alcohol extract groups. Second, lipid vacuoles were increased with the concomitant decrease in the number of hepatopancreatic cell nuclei in the FB and FB alcohol extract groups. Alcohol extract from clove basil leaf also contains an unidentified substance(s) that affects fish lipid metabolism

(*Abdel-Tawwab et al., 2018*), although its effect is opposite (i.e., reduces fat accumulation), reflecting the diversity of plant-derived substances.

In fish, alkaline phosphatase (AKP), glutamic oxalacetic transaminase (GOT) and glutamic-pyruvic transaminase (GPT) are key enzymes in amino acid metabolism and have been used as important indices to evaluate hepatopancreatic conditions (*Wu et al., 2017*). Higher activities of these enzymes in the hepatopancreas generally imply that fish are under stressful conditions (*Wang et al., 2014*). These are also deviation enzymes released into plasma upon hepatopancreas damage and dysfunction (*Li et al., 2017*; *Tan et al., 2017*). The activities of these enzymes in the hepatopancreas fluctuated in the present study but were generally comparable to those of the control group. The serum activities of AKP, GOT, and GPT were sometimes higher than those of the control group, suggesting the hepatopancreatic damage. Combined with the hepatopancreas histology, it is speculated that the FB water extract induced minimal hepatopancreas damage, the FB alcohol extract induced mild hepatopancreas damage, and the whole FB caused the most serious hepatopancreas damage.

Furthermore, oxidative parameters indicated that FB residues induced hepatopancreas damage more than other extracts. The $NADP^+$, T-GSH and GSSH contents of the FB and FB residues groups were higher than those of the control group. The histology of the FB residues group also showed serious hepatopancreas damage. NADPH plays an important role in protecting cells from oxidative stress, since it affects the content of glutathione (T-GSH, GSSH), which reduces the level of $H_2O_2$ (*Boonyuen et al., 2017*; *Cappellini & Fiorelli, 2008*). It has been found that faba bean cause hepatopancreas damage and reduce nonspecific immune response of tilapia (*Chen, 2015*), but it is unclear which component of the faba bean causes the damage. Further studies on FB residues will help the identification of the oxidative compounds in faba bean, possibly facilitating the effective removal of the oxidative substances.

A limitation of this study is that the experimental time lasted for 100 d through fall and winter. Although the water temperature was kept relatively constant (25–30 °C), the activities of metabolic enzymes are influenced by water temperature and seasonal variation (*Qiang et al., 2014*; *Jobling, 2016*; *Richard et al., 2016*). Our experimental conditions should have practical implications for the application of faba bean in aquaculture, but this may explain why some enzyme activities and tissue weights were not consistent between 50 d and 100 d. In this regard, liver histology may provide more robust information about hepatopancreatic damages rather than enzymatic activities during the experimental period. Another limitation is that the diets were not isoenergetic or isonitrogenous, which may be related to the inconsistency between nutrition and health status. In particular, the four FB extracts differed in proteins content, which affects fish physiology to a great extent. Although our main finding was based on the comparison between the FB water extract and commercial feed that have similar protein content, feed formulation requires improvements to gain mechanical insights. In the near future, we will further examine the effects of these four FB extracts by formulating isoenergetic and isonitrogenous diets containing faba bean extracts.

## CONCLUSION

In the present study, the FB water extract group had a better health status than FB and other extract groups in terms of rapid growth, normal lipid metabolism, and minimal hepatopancreas damage. The FB alcohol extract and proteins groups had an inferior health status compared with the FB water extract group, with the FB and FB residues groups ranking last. Therefore, this study successfully separated the beneficial and detrimental substances of faba bean. Given that the FB water extract by itself can improve the textural quality (*Ma et al., 2020*), this could become a promising food additive in grass carp culture. In the near future, we will further analyze the beneficial and detrimental effects of these four FB extracts by formulating isoenergetic and isonitrogenous diets, and carry out a detailed evaluation based on intestinal microbes and metabolites, etc.

### Funding

This study was funded by the Modern Agro-industry Technology Research System (No. CARS-45-21). The funders had no role in study design, data collection and analysis, decision to publish, or preparation of the manuscript.

### Grant Disclosures

The following grant information was disclosed by the authors:
Modern Agro-industry Technology Research System: CARS-45-21.

### Competing Interests

The authors declare that they have no competing interests.

### Author Contributions

- Lingling Ma conceived and designed the experiments, performed the experiments, analyzed the data, prepared figures and/or tables, authored or reviewed drafts of the paper, and approved the final draft.
- Gen Kaneko conceived and designed the experiments, authored or reviewed drafts of the paper, and approved the final draft.
- Jun Xie conceived and designed the experiments, authored or reviewed drafts of the paper, and approved the final draft.
- Guangjun Wang conceived and designed the experiments, authored or reviewed drafts of the paper, and approved the final draft.
- Zhifei Li performed the experiments, authored or reviewed drafts of the paper, and approved the final draft.
- Jingjing Tian performed the experiments, authored or reviewed drafts of the paper, and approved the final draft.
- Kai Zhang performed the experiments, authored or reviewed drafts of the paper, and approved the final draft.
- Yun Xia analyzed the data, prepared figures and/or tables, and approved the final draft.

- Wangbao Gong analyzed the data, prepared figures and/or tables, and approved the final draft.
- Haihang Li conceived and designed the experiments, authored or reviewed drafts of the paper, and approved the final draft.
- Ermeng Yu conceived and designed the experiments, analyzed the data, prepared figures and/or tables, authored or reviewed drafts of the paper, and approved the final draft.

## Animal Ethics

The following information was supplied relating to ethical approvals (i.e., approving body and any reference numbers):

The Animal Ethics Committee of the Guangdong Provincial Zoological Society, China, approved this study under permit number GSZ-AW012.

## Data Availability

The raw measurements are available in the Supplemental Files.

## Supplemental Information

Supplemental information for this article can be found online at http://dx.doi.org/10.7717/peerj.9516#supplemental-information.

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
