# Peer review of "Safety evaluation of four faba bean extracts used as dietary supplements in grass carp culture based on hematological indices, hepatopancreatic function and nutritional condition"

_PeerJ, doi:10.7717/peerj.9516_

## Round 0.1 · original submission · Major Revisions

Your manuscript needs a deep revision. Many data about the diet used in this work need to be added in materials and methods because these data are crucial for the work. Other comments have been suggested by the reviewers.

Reviewer 1 ·

Basic reporting

faba bean was fed to big grass carp for improve the muscle texture, the fish is named with crisp grass carp. the mn is helpful to understand the effects of faba bean on grass carp. the results is benifical to the crisp grass carp production.

Experimental design

why the experimental diets are very different protein content?

Validity of the findings

no comment.

Additional comments

1.why the experimental diets are very different protein content? the protein content of FBA and FBW is very higher than control group, the results indicated FBW can improve the growth performance, we can conclude the reason is the higher dietary protein intake. why FBW protein is so higher? Can you supply the composition of FBW,FBA?
2.information of the commercial diet need to be supplied.
3.whteher the growth performance is same to the published paper(Effects of four faba bean extracts on growth parameters, textural quality, oxidative responses, and gut characteristics in grass carp,Aquaculture,2020,https://doi.org/10.1016/j.aquaculture.2019.734620)? but the resutes of growth is not consistent.you should give me a reasonable explanation.

Reviewer 2 ·

Basic reporting

it is very difficult to read the data in Table 3 due to the format

Experimental design

no comment

Validity of the findings

no comment

Additional comments

In this manuscript, four FB extracts (water extract, alcohol extract, proteins, and residues) were mixed with commercial feed and fed to grass carp (Ctenopharyngodon idellus) using whole FB and commercial feed as controls and examined their beneficial and detrimental effects for the safety evaluation. It is a worthy work. However, there are some issues that need to be clarified.
Main concerns:

Line 90, preparation of faba bean extracts should be briefly introduced and their corresponding contents in different diets should be added.

Line 91, the author referred that 16 kg was used for each of the four experimental diets (each for 16 kg), it might be not very reasonable due to not enough diets to feed all sample fish. In the fish culture, grass carp with a body weight of 750 ± 50 g were randomly allocated into six groups with two tanks per group, and each tank was maintained at a density of eight individuals. Fish were fed for 100 days and the feed amount of each day was 3%-5% of fish weight. And on 50 d, three fish were randomly taken from each group. Therefore, for the diet of each group, it was 0.75*2*8*(3%-5%)*100-0.75*3*(3%-5%)*50≈43.5 kg at least.

Line 166-167, the author referred that four FB extract groups had markedly higher hepatopancreatic weight than the FB and control groups on 100 d (P < 0.05), it is not accurate as FBP extract groups (19.12±4.64ab) had not markedly higher hepatopancreatic weight than the FB groups (9.82±2.35b) and the control group (10.95±3.66b) on 100 d. And please give an explanation why the intestinal weight of the control group was significantly lower than those of other groups (P < 0.05) on 100 d, and why many of the hepatopancreas, Visceral fat and Intestinal weight in different groups on 100 d were reduced compared to 50 d, although a limitation of this study were given that the experimental time lasted for 100 d through fall and winter in the discussion.

Line 172-183, in the hematological analyses, it is very difficult to read the data in Table 3 due to the format, which needs to be improved. It is necessary to explain why these hematological parameters such as red blood cell (RBC), hematocrit (HCT) and hemoglobin (HGB) contents were chosen. Furthermore, many results in the hematological analyses were questionable. For example, the author referred that clear differences were observed on 100 d with small deviations, where FB and FB water extract groups showed lower counts of red blood cell (RBC), hematocrit (HCT) and hemoglobin (HGB) contents, some of which were statistically significant (P < 0.05), but in fact only FB water extract groups showed significant difference in RBC.

Line 205-206, in Fig 3, total cholesterol content (TC) levels were markedly low on 50 d with no significant differences between groups, but increased by 100 d, which is very large difference and change, a reasonable explanation should be provided.

Minor comments:

The location of Fig 5B and C was missing.
The format of references should be consistent, e.g. line 344 and 353 DOI did not need to change a line; line 389, DOI should not be bold.

Annotated reviews are not available for download in order to protect the identity of reviewers who chose to remain anonymous.

---

## Round 0.2 · Minor Revisions

Your manuscript still has to be improved due to some minor concerns. One of them is to clarify the protein difference of experimental diets. Other minor comments are suggested by reviewer 1.

Reviewer 1 ·

Basic reporting

no comment。

Experimental design

no comment

Validity of the findings

no comment

Additional comments

The revision of mn had been improved by the author, but i still had some question
1.about protein difference of experimental diets,whether FB water and alcohol extracts improved growth performance? because the high protein,we cant make a conclusion in the abstract.
2.such as the new reference should be cited by the mn,
Xiaoxia Li, Shijun Chen, Jijia Sun et al. Partial substitution of soybean meal with faba bean meal in grass carp (Ctenopharyngodon idella) diets, and the effects on muscle fatty acid composition, flesh quality, and expression of myogenic regulatory factors. Journal of the World Aquaculture Society. 2019.
LianGan, Yan-zhiWang, Shi-junChen et al. Identification and characterization of long non-coding RNAs in muscle sclerosis of grass carp, Ctenopharyngodon idellus fed with faba bean meal. Aquaculture, 2020,516:734521.
Wei-HuaXu, Hong-HongGuo, Shi-JunChenet al. Transcriptome analysis revealed changes of multiple genes involved in muscle hardness in grass carp (Ctenopharyngodon idellus) fed with faba bean meal. Food chemistry,2020, 314:126205.
Proteomic and metabolomic basis for improved textural quality in crisp grass carp (Ctenopharyngodon idellus C.et V) fed with a natural dietary pro-oxidant.Food Chemistry ,2020,325:126906.

---

## Round 0.3 · accepted · Accept

Dear Authors,

I am pleased to confirm that your paper has been accepted for publication in PeerJ.

Thank you for submitting your work to this journal.